# The diagnostic value of decreased levels of inflammatory markers for the state of hepatitis C virus (HCV) infection

Lu Han, Lei Bi, Xinpeng Li 🄳 *

Department of Clinical Laboratory, Public Health Clinical Center of Chengdu, Chengdu, China

* lxpapril@outlook.com

## Abstract

### Background

Blood-cell-based inflammatory biomarkers are increasingly recognized for their diagnostic value in infections due to their clinical accessibility. With Hepatitis C virus (HCV) incidence rising and its often asymptomatic onset, this study aims to improve diagnostic evidence for HCV by analyzing changes in these biomarkers.

### Methods

Utilizing NHANES database, we employed binary logistic regression and generalized additive models to explore the relationship between systemic inflammatory index and HCV infection. Three adjusted models controlled for confounders, and subgroup analyses were stratified by age, gender, race, and BMI.

### Results

Significant differences were observed in PLR (103.24±44.59), SII (455.23±339.56), PNR (58.22±32.20), PMR (366.85±191.76), and NMR (7.03±3.78) between infected and uninfected groups (P<0.05). Adjusted analyses revealed associations between anti-HCV and Log2-PLR (OR = 0.58), Log2-SII (OR = 0.64), Log2-PMR (OR = 0.77), and Log2-NMR (OR = 0.79). Individuals under 30 showed no significant differences. A unit increase below 9.30 in Log2-PMR reduced HCV risk by 0.60-fold. PMR demonstrated an AUC of 0.648, specificity 0.7632, and sensitivity 0.4709.

### Conclusion

In individuals aged 30 and above, inflammatory markers PLR, SII, PMR, and NMR decrease in HCV cases. Variability across races, genders, and BMI groups highlights their diagnostic utility in diverse populations.

**Data availability statement:** The original datasets, including demographic and laboratory data, were downloaded from the official NHANES website (https://wwwn.cdc.gov/nchs/nhanes/search/datapage.aspx?Component=Laboratory&CycleBeginYear=2017), and subsequently merged for the years 2007-2018 for analysis. The merged data set is available at the following link: https://doi.org/10.6084/m9.figshare.29565611. The full data used in the analysis is also provided in supporting information.

**Funding:** The author(s) received no specific funding for this work.

**Competing interests:** The authors have declared that no competing interests exist.

**Abbreviations:** ALB, Albumin; ALT, Alanine aminotransferase; AST, Aspartate aminotransferase; BMI, Body mass index; CI, Confidential interval; GGT, Gamma glutamyl transferase; HCC, Hepatocellular carcinoma; HCV, Hepatitis C virus; LMR, Lymphocyte to monocyte ratio; NHANES, National Health and Nutrition Examination Survey; NLR, Neutrophil to lymphocyte ratio; NMR, Neutrophil to monocyte ratio; OR, Odds rations; PBMC, Peripheral blood mononuclear cells; PIR, Poverty income ratio; PLR, Platelet to lymphocyte ratio; PMR, Platelet to monocytes ratio; PNR, Platelet to neutrophil ratio; ROC, Receiver operating characteristic; SII, Systemic immune inflammatory index; SIRI, Systemic inflammatory response index.

# 1 Introduction

Hepatitis C, attributed to the hepatitis C virus (HCV), emerged as a documented viral hepatitic illness in 1989 [1]. The transmission of HCV primarily occurs through diverse modalities, including sexual contact, vertical transmission from mother to infant, blood transfusion, and intravenous drug use. Globally prevalent, approximately 578 million individuals were affected by chronic HCV infection as of 2019. with an annual report of 1.5 million new infections [2]. Notably, from 2015 to 2020, the number of individuals undergoing treatment for chronic HCV infection witnessed a tenfold increase [3]. Hepatitis C has become a significant public health concern, jeopardizing both the health and quality of life of affected individuals [4]. Infection with HCV can manifest as asymptomatic or present with mild symptoms such as fatigue, nausea, loss of appetite, and abdominal pain. Among patients, around 15% exhibit acute symptoms, which include liver pain, jaundice (yellow discoloration of the skin and eyes), and long-term liver damage while approximately 80% of those exposed to the HCV progress to chronic hepatitis C [4]. In case of chronic inflammation, necrosis, and fibrosis of the liver, some patients may advance to cirrhosis or even hepatocellular carcinoma [5]. Beyond hepatic damage and the elevated risk of cirrhosis and hepatocellular carcinoma (HCC), chronic HCV infection can induce persistent inflammation. This is evidenced by qualitative and quantitative changes in the immune repertoire and tissue microenvironment, subsequently leading to various tumors. Examples include cholangiocarcinoma, pancreatic cancer, and non-Hodgkin lymphoma associated with chronic immune stimulation [6–8].

The blood-cell-based inflammatory biomarkers, including neutrophil to lymphocyte ratio (NLR), lymphocyte to monocyte ratio (LMR), platelet to lymphocyte ratio (PLR), platelet to neutrophil ratio (PNR), neutrophil to monocyte ratio (NMR), platelet to monocytes ratio (PMR), systemic immune inflammatory index (SII) and systemic inflammatory response index (SIRI) [9,10], are typically calculated using the peripheral blood count. They are often compared with other inflammation markers due to their accessibility and cost-effectiveness. Numerous studies have investigated the diagnostic efficacy of routine blood-related ratio in various diseases [11,12]. For instance, a study demonstrated that NLR, PNR, MLR and NMR as potential biomarkers for predicting lupus nephritis (LN). NLR and MLR proved effective in distinguishing between LN patients without infection from healthy subjects. Furthermore, NLR, NMR, and PNR were found to be valuable in differentiating LN patients with infection and flare [13]. Another investigation revealed a significantly higher PLR in ankylosing spondylitis (AS) patients with a bath ankylosing spondylitis disease activity index (BASDAI) score ≥ 2. Additionally, patients with higher PLR reported greater pain level [14]. In a ten year follow up study in 85,154 individuals, results indicated that SII and SIRI increased the risk of stroke, two stroke subtypes, and all-cause death. Higher SIRI was associated with increased incidence of myocardial infarction (MI) [15]. Similar findings were corroborated in another large-scale study [16].

Regarding these blood-related inflammatory markers, there is also a strong association with liver disease. A study on the mortality of HBV related liver cirrhosis showed

that LMR is an independent predictive factor of mortality [17]. Another study focusing on the treatment of HCC revealed that NLR is a reliable and cost-effective biomarker, assisting in determining outcomes following HCC treatment [18]. In the context of HCV infection, several studies highlight the effectiveness of these ratios. For example, PLR is closely related to disease severity in patients with HCV related liver disease [19]. In patients with chronic hepatitis C whose serum viral RNA is repeatedly negative, residual HCV RNA after antiviral treatment may persist in liver tissue and peripheral blood mononuclear cells (PBMC) for an extended duration [20,21]. One study demonstrated that a significantly higher NLR 24 weeks after the start of treatment predicted the elimination of replicative HCV RNA strands [22].

All of the above-mentioned studies indicate that these inflammation indicators based on blood routine examinations hold certain value for latent diseases, including the viral proliferation status of viral diseases.Therefore, the aim of this article is to conduct a thorough investigate into the association between HCV infection and significant inflammatory markers. This will be accomplished by utilizing well-established studies and population-based data from the NHANES official data, offering novel perspectives and methodologies for the early diagnosis and treatment of liver diseases.

## 2 Methods

### 2.1 HCV Diagnosis

NHANES utilizes the COBAS AMPLICOR HCV MONITOR Test, version 2.0 (v2.0), an in vitro nucleic acid amplification test, to quantify HCV RNA in human serum or plasma on the COBAS AMPLICOR Analyzer. Subsequently, the participants were categorized into two groups based on the laboratory examination results: anti-HCV (+) individuals with HCV infection and anti-HCV (-) individuals without infection.

### 2.2 Detecting HCV antibody and routine blood-cell-related inflammatory markers

HCV antibody was tested based on the VITROS Anti-HCV Reagent Pack and VITROS Immunodiagnostic Products Anti-HCV Calibrator on the VITROS ECi/ECiQ Immunodiagnostic Systems, the VITROS 3600 Immunodiagnostic System and the VITROS 5600 Integrated System. This involves a two-stage reaction. In the first stage, HCV antibody present in the sample binds with HCV recombinant antigens coated on the wells. Unbound sample is removed by washing. In the second stage, horseradish peroxidase (HRP) – labeled antibody conjugate (mouse monoclonal anti-human IgG) binds to any human IgG captured on the well in the first stage. Unbound conjugate is removed by washing. The system signal intensity is measured by chemiluminescence, and the signal intensity is directly proportional to the anti-HCV level in the sample.

The ratios of NLR, LMR, PLR, PNR, PMR and NMR mainly depend on the results obtained from a complete blood count. The NHANES five-part differential CBC utilizes the Beckman-Coulter method, which employs VCS techniques for counting and scaling using. The test includes automated sample dilution, mixing and hemoglobin measurement using a single-beam photometer. Furthermore, platelet count (PC), neutrophil count (NC), lymphocyte count (LC), and monocyte count (MC) are used in the calculation of these ratios. For instance, NLR is determined as NC/LC (neutrophil to lymphocyte ratio), LMR represents lymphocyte to monocyte ratio, PLR stands for platelet to lymphocyte ratio, PNR is platelet to neutrophil ratio, PMR is platelet to monocytes ratio, and NMR is neutrophil to monocyte ratio. Additionally, the SII is defined as (platelets × neutrophils)/ lymphocytes, while the SIRI is defined as (neutrophils × monocytes)/lymphocytes.

### 2.3 Study population

We conducted a cross-sectional observational study using data from the National Health and Nutrition Examination Survey (NHANES), a nationally representative survey by the Centers for Disease Control and Prevention (CDC) in the United States (https://www.cdc.gov/nchs/nhanes/index.htm). NHANES is designed to provide a nationally representative assessment of the health and nutritional status of the U.S. population. Our study encompassed six consecutive surveys cycles: 2007–2008, 2009–2010, 2011–2012, 2013–2014, 2015–2016, and 2017–2018.

Data, including demographic information, examination results and laboratory findings were collected from the NHANES participants (59842). Exclusion criteria were applied as follows: individuals with missing data on peripheral blood cell count (10705) and HCV antibody test (27499) were excluded. Cases testing positive for HIV antibody (34) and HCG (166) were removed to eliminate potential interference from immune factors. Additionally, 3921 participants with null value of covariates were excluded. Ultimately, a total of 17526 of 59842 participants were enrolled in the analysis (Fig 1).

## 2.4 Covariates

To assess factors potentially associated with HCV antibody presence, the following covariates were incorporated: gender, age, race/ethnicity, body mass index (BMI), poverty income ratio (PIR), diabetes, and liver function makers. Age was stratified into three stages: <30 years, 30–60 years, >60 years. Race and ethnicity were grouped as Mexican American and other Hispanic, Non-Hispanic White and Other race – Including Multi-Racial, Non-Hispanic Black, and Non-Hispanic Asian. BMI was classified as normal weight (< 25 kg/m$^2$), overweight (≥ 25 kg/m$^2$ and < 30 kg/m$^2$) and obesity (≥ 30 kg/m$^2$). Diabetes was defined based on the affirmative response to the question "Doctor told you have diabetes". Liver function was evaluated using alanine aminotransferase (ALT), aspartate aminotransferase (AST), gamma glutamyl transferase (GGT), and albumin (ALB).

## 2.5 Statistical analysis

Statistical analyses were conducted following the survey guidelines provided by NHANES. Continuous variables were expressed as the means with standard deviation (Mean ± SD), while categorical variables were presented as absolute values (percentages). Binary logistic regression models were established to assess odds rations (OR) and 95% confidential interval (CI) for the association between systemic inflammation biomarkers and anti-HCV. Three models were used to ensure model stability: model 1 (unadjusted), model 2 (adjusted for age, gender and race/ethnicity), and model 3 (fully adjusted). Subgroup analyses, stratified by gender, age, race/ethnicity, and BMI, were performed. Generalized additive

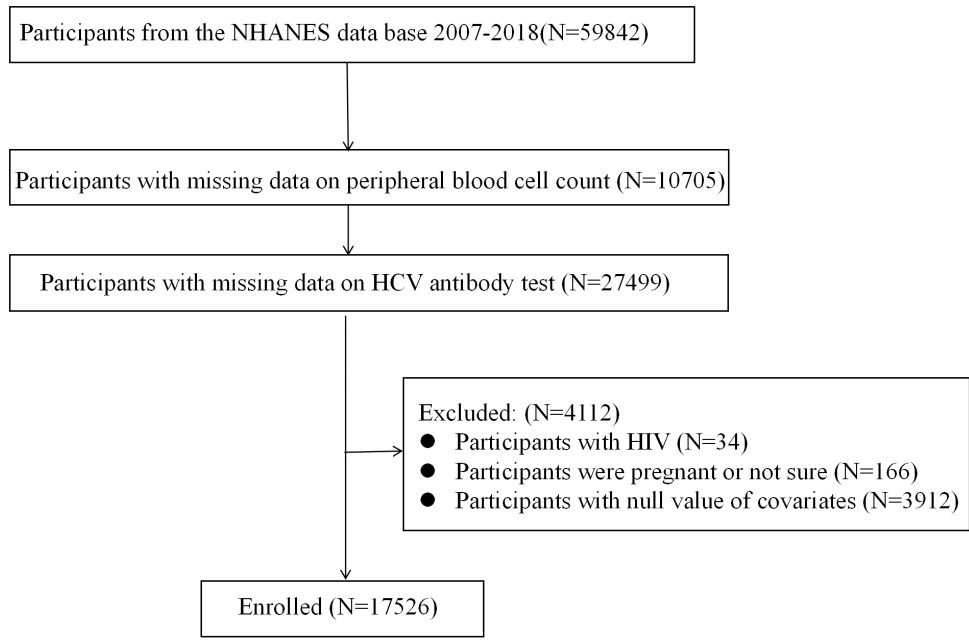

**Fig 1. Exclusion and enrollment summary from NHANES.**

models were utilized to simulate the nonlinear relationship between the indices and anti-HCV. The inflection point was calculated upon non-linearity using a recursive algorithm. Receiver operating characteristic (ROC) curves were generated to predict the sensitivity and specificity of the biomarkers and anti-HCV. All estimates, accounting for NHANES, incorporated calculated sample weights. The analyses were executed using R 4.3.0 (https://www.r-project.org/) and EmpowerStats 4.1 (https://www.empowerstats.net/cn/index.php). A significance level of $P < 0.05$ was considered statistically significant.

## 2.6 Ethics approval and consent to participate

The data was obtained from a publicly available database. The initial survey protocol underwent thorough ethical examination and obtained approval from the Institutional Review Board of the Centers for Disease Control and Prevention. The studies adhered to local legislation and institutional requirements. Participants' legal guardians or next of kin provided written informed consent for their participation in this study.

# 3 Results

## 3.1 Characteristics of participants

All participants (17526) were categorized into anti-HCV negative (17165) and anti-HCV positive (361) groups (Table 1). Among the anti-HCV positive population, males accounted for 72%, with the majority falling within the age of 30–60 years (80.21%).

**Table 1. Demographic information and baseline population characteristics.**

| | Total | anti-HCV Negative | anti-HCV Positive | P value |
|---|---|---|---|---|
| **Number** | 17526 | 17165 | 361 | |
| **Gender (%)** | | | | <0.0001 |
| Male | 49.05 | 48.66 | 72.00 | |
| Female | 50.95 | 51.34 | 28.00 | |
| **Age (Years, %)** | | | | <0.0001 |
| <30 | 27.63 | 28.07 | 2.47 | |
| 30≤Age≤60 | 48.92 | 48.38 | 80.21 | |
| >60 | 23.44 | 23.55 | 17.32 | |
| **Race/ethnicity (%)** | | | | <0.0001 |
| Mexican American and Hispanic | 16.30 | 16.40 | 10.17 | |
| Non-Hispanic White and Multi-Racial | 67.38 | 67.36 | 68.96 | |
| Non-Hispanic Black | 11.03 | 10.88 | 20.14 | |
| Non-Hispanic Asian | 5.28 | 5.36 | 0.73 | |
| **BMI (%)** | | | | 0.0016 |
| <25 | 31.96 | 31.81 | 40.10 | |
| 25≤BMI<30 | 31.27 | 31.25 | 32.05 | |
| ≥30 | 36.78 | 36.93 | 27.86 | |
| **PIR** | 3.10±1.68 | 3.12±1.68 | 2.01±1.58 | <0.0001 |
| **Diabetes (%)** | | | | 0.5758 |
| Yes | 9.99 | 9.98 | 10.66 | |
| No | 87.74 | 87.76 | 86.17 | |
| Unkown | 2.27 | 2.26 | 3.17 | |
| **ALT (U/L)** | 24.26±20.33 | 23.55±16.74 | 65.72±79.81 | <0.0001 |
| **ALB (g/dL)** | 4.26±0.35 | 4.27±0.35 | 4.08±0.38 | <0.0001 |
| **AST (U/L)** | 24.51±15.91 | 23.89±13.95 | 60.69±48.09 | <0.0001 |
| **GGT (IU/L)** | 27.14±39.55 | 26.22±36.77 | 80.79±104.76 | <0.0001 |

Non-Hispanic Asians exhibited a lower anti-HCV positive rate (0.73%), while the proportion was slightly higher among Non-Hispanic Blacks (20.14%). Individuals infected HCV tended to display a predisposition towards a lean body composition (BMI<25, 40.1%) and a lower PIR (2.01±1.58). Liver function markers, including ALT (65.72±79.81), AST (60.69±48.09) and GGT (80.79±104.76) were significantly higher in individuals infected with hepatitis C compared uninfected.

Systemic inflammatory markers such as PLR (103.24±44.59), SII (455.23±339.56), PNR (58.22±32.20), PMR (366.85±191.76), and NMR (7.03±3.78) were found to be lower in the population infected with HCV compared to the normal population ($P < 0.05$) (Table 2).

### 3.2 Relationship between systemic inflammation biomarkers and hepatitis C

The associations between HCV infection and PLR, along with other inflammatory markers, are summarized in Table 3. In the absence of covariates adjustments (Model 1), and with adjustments solely for age, gender, and race/ethnicity (Model 2), all markers, exhibited an inverse association with HCV infection. However, following comprehensive covariates adjustments, PNR (OR= 0.97, 95% CI: 0.80, 1.17) ($P = 0.7504$) failed to demonstrate a significant association with HCV infection. Conversely, PLR (OR= 0.58, 95% CI: 0.49, 0.68), SII (OR= 0.64, 95% CI: 0.56, 0.73), PMR (OR= 0.77, 95% CI: 0.63, 0.95), NMR (OR= 0.79, 95% CI: 0.64, 0.98) displayed negative associations with hepatitis C.

### 3.3 Stratified analysis between PLR, SII, PMR and NMR with hepatitis C

As showed in Table 4, we employ further stratification based on four variables, gender, age, race, and BMI with the aim of elucidating the relationship between hepatitis C and blood-related ratios across diverse demographic groups. PLR (OR

**Table 2. Comparison of systemic inflammation biomarkers in infected and uninfected populations.**

|  | anti-HCV Negative | anti-HCV Positive | *P* value |
|---|---|---|---|
| NLR | 2.13±1.13 | 2.09±1.29 | 0.5633 |
| PLR | 120.01±44.24 | 103.24±44.59 | <0.0001 |
| SII | 514.38±305.56 | 455.23±339.56 | 0.001 |
| SIRI | 1.29±0.92 | 1.35±1.00 | 0.2592 |
| PNR | 63.73±27.99 | 58.22±32.20 | 0.0008 |
| LMR | 4.00±1.79 | 3.81±1.59 | 0.0641 |
| PMR | 450.04±174.68 | 366.85±191.76 | <0.0001 |
| NMR | 7.62±2.87 | 7.03±3.78 | 0.0005 |

Kruskal–Wallis test was used for comparisons of continuous variables.

**Table 3. Exploring the correlation between systemic inflammation biomarkers and HCV infection (logarithm base 2).**

| Exposure | Model 1 | Model 2 | Model 3 |
|---|---|---|---|
| Log PLR | 0.50 (0.42, 0.58) <0.0001 | 0.55 (0.47, 0.64) <0.0001 | 0.58 (0.49, 0.68) <0.0001 |
| Log SII | 0.53 (0.47, 0.60) <0.0001 | 0.59 (0.52, 0.66) <0.0001 | 0.64 (0.56, 0.73) <0.0001 |
| Log PNR | 0.75 (0.64, 0.89) 0.0012 | 0.83 (0.70, 0.99) 0.0427 | 0.97 (0.80, 1.17) 0.7504 |
| Log PMR | 0.40 (0.34, 0.48) <0.0001 | 0.54 (0.45, 0.65) <0.0001 | 0.77 (0.63, 0.95) 0.0135 |
| Log NMR | 0.48 (0.40, 0.58) <0.0001 | 0.64 (0.53, 0.78) <0.0001 | 0.79 (0.64, 0.98) 0.0316 |

Model 1 adjusted no variables.

Model 2 adjusted age, gender, and race/ethnicity.

Model 3 adjusted age, gender, race/ethnicity, BMI, PIR, diabetes, ALT, ALB, AST, and GGT.

**Table 4. Stratified associations between systemic inflammation biomarkers and hepatitis C (OR 95%, *P* value, LOG base 2).**

| | Log PLR | Log SII | Log PMR | Log NMR |
|---|---|---|---|---|
| **Gender** | | | | |
| Male | 0.54 (0.44, 0.65) <0.0001 | 0.61 (0.52, 0.70) <0.0001 | 0.43 (0.35, 0.53) <0.0001 | 0.58 (0.46, 0.73) <0.0001 |
| Female | 0.48 (0.36, 0.64) <0.0001 | 0.45 (0.36, 0.56) <0.0001 | 0.51 (0.36, 0.70) <0.0001 | 0.45 (0.32, 0.62) <0.0001 |
| **Age** | | | | |
| <30 | 0.67 (0.12, 3.87) 0.6543 | 0.86 (0.30, 2.50) 0.7851 | 1.63 (0.35, 7.66) 0.5326 | 1.96 (0.41, 9.30) 0.3965 |
| 30≤Age≤60 | 0.40 (0.31, 0.50) <0.0001 | 0.49 (0.42, 0.58) <0.0001 | 0.31 (0.25, 0.39) <0.0001 | 0.38 (0.30, 0.47) <0.0001 |
| >60 | 0.60 (0.49, 0.74) <0.0001 | 0.53 (0.44, 0.64) <0.0001 | 0.63 (0.47, 0.85) 0.0029 | 0.52 (0.37, 0.73) 0.0002 |
| **Race** | | | | |
| Mexican American and Hispanic | 0.38 (0.24, 0.62) 0.0001 | 0.33 (0.24, 0.46) <0.0001 | 0.34 (0.22, 0.53) <0.0001 | 0.32 (0.20, 0.49) <0.0001 |
| Non-Hispanic White and Multi-Racial | 0.57 (0.46, 0.72) <0.0001 | 0.72 (0.59, 0.88) 0.0013 | 0.45 (0.35, 0.58) <0.0001 | 0.82 (0.60, 1.13) 0.2266 |
| Non-Hispanic Black | 0.45 (0.34, 0.59) <0.0001 | 0.55 (0.45, 0.67) <0.0001 | 0.40 (0.31, 0.52) <0.0001 | 0.54 (0.41, 0.72) <0.0001 |
| Non-Hispanic Asian | 0.48 (0.18, 1.30) 0.1486 | 0.47 (0.16, 1.43) 0.1850 | 0.39 (0.19, 0.81) 0.0123 | 0.41 (0.07, 2.61) 0.3490 |
| **BMI** | | | | |
| <25 | 0.46 (0.35, 0.60) <0.0001 | 0.61 (0.49, 0.76) <0.0001 | 0.41 (0.31, 0.54) <0.0001 | 0.60 (0.44, 0.83) 0.0016 |
| 25≤BMI<30 | 0.57 (0.44, 0.73) <0.0001 | 0.60 (0.49, 0.74) <0.0001 | 0.43 (0.32, 0.57) <0.0001 | 0.55 (0.40, 0.76) 0.0004 |
| ≥30 | 0.45 (0.33, 0.61) <0.0001 | 0.40 (0.32, 0.49) <0.0001 | 0.37 (0.27, 0.50) <0.0001 | 0.34 (0.25, 0.47) <0.0001 |

0.54; 95% CI 0.44–0.65), SII (OR 0.61; 95% CI 0.52–0.70), and NMR (OR 0.58; 95% CI 0.46–0.73), generally exhibit higher values in males compared to females, except for PMR (OR 0.43; 95% CI 0.35–0.53). Among individuals below the age of 30, all blood-related ratios show no statistically significant differences (*P*<0.05). Noteworthy statistical significance is observed exclusively in PMR values across demographic groups, including Mexican American and Hispanic (OR 0.34; 95% CI 0.22–0.53), Non-Hispanic White and Multi-Racial (OR 0.45; 95% CI 0.35–0.58), Non-Hispanic Black (OR 0.40; 95% CI 0.31–0.52), and Non-Hispanic Asian (OR 0.34; 95% CI 0.25–0.47). A consistent decreasing trend is observed with an increase in BMI, evident in both SII [BMI<25 kg m-2 (OR 0.61; 95% CI 0.49–0.76), ≥25 and < 30 kg m-2 (OR 0.60; 95% CI 0.49–0.74), ≥30 kg m-2 (OR 0.40; 95% CI 0.32–1.0.49)] and NMR [BMI<25 kg m-2 (OR 0.60; 95% CI 0.44–0.83), ≥25 and < 30 kg m-2 (OR 0.55; 95% CI 0.40–0.76), ≥30 kg m-2 (OR 0.34; 95% CI 0.25–0.47)].

### 3.4 Dose-response relationship between PLR, SII, PMR, NMR and anti-HCV

The generalized additive model was utilized to investigate the dose-response relationship between hepatitis C and systemic inflammatory factors (PLR, SII, PMR, NMR) with a base of log2 (Fig 2). Consistent with the outcomes in Table 1, individuals with lower levels of inflammatory factors tend to lean towards a positive anti-HCV result. As the numerical values of PLR (Fig 2A), SII (Fig 2B), PMR (Fig 2C), and NMR (Fig 2D) increase, the anti-HCV outcome gradually tends towards negativity (where 1 signifies positive and 0 signifies negative). The log-likelihood ratio test was employed to compare a non-segmented model with a segmented regression model, aiming to determine the presence of a threshold. The inflection point connecting the segments was derived from the maximum likelihood model, determined through a two-step recursive method. Taking PMR as an example (Fig 2C), the calculated inflection point (K) is 9.30. In theory, when the Log2-PMR value is less than 9.30, for each unit increase in Log PMR, the positive risk of HCV decreases by 0.60 times (1 minus the OR 0.40) (Table 5).

### 3.5 Diagnostic value of certain indicators based on ROC curves

The AUC values varied among different markers: PMR (AUC=0.648) with a specificity of 0.7632 and sensitivity of 0.4709; SII (AUC=0.643) with a specificity of 0.8046 and sensitivity of 0.4349; PLR (AUC=0.623) with a specificity of 0.6822 and

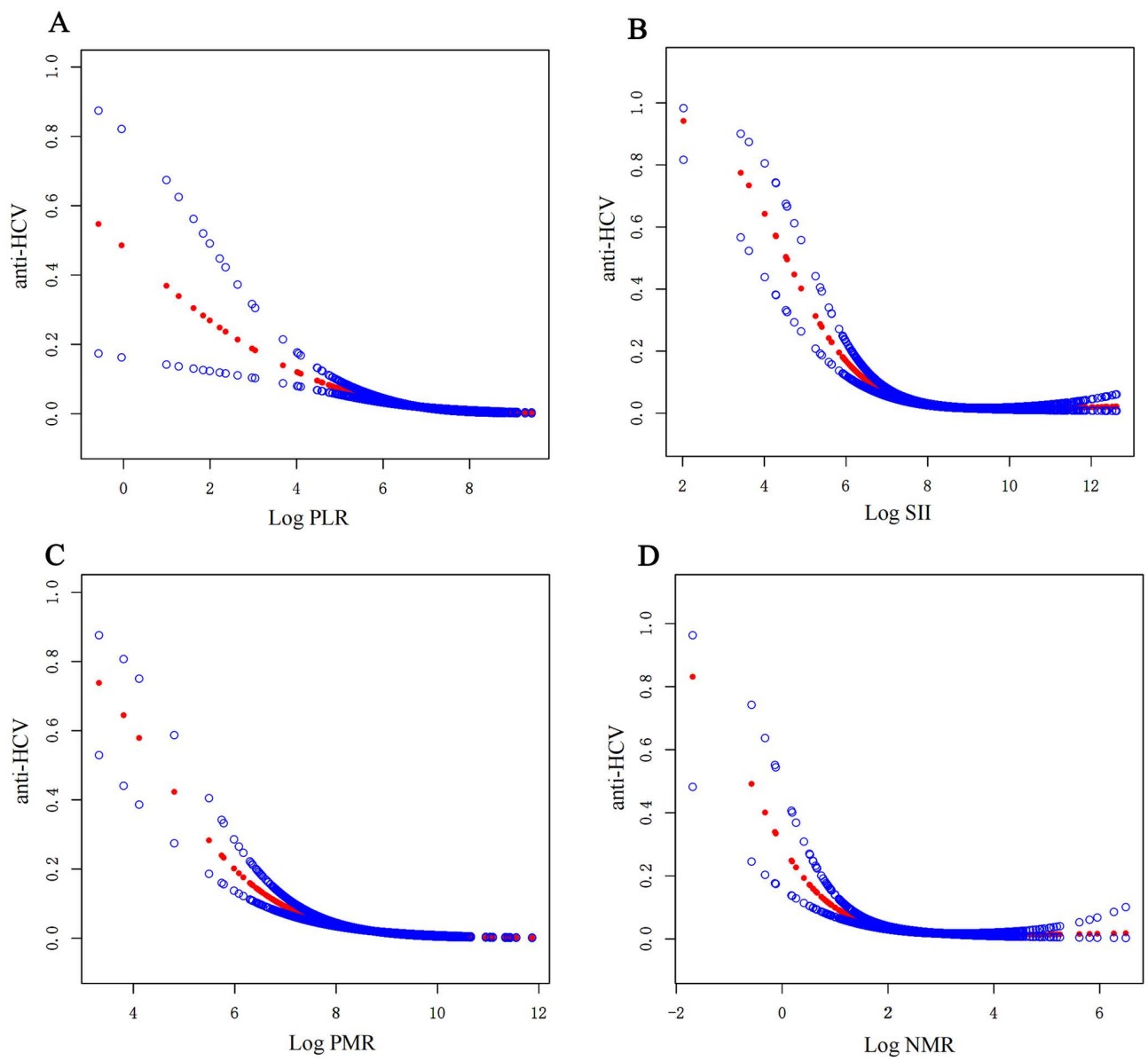

**Fig 2. General addictive models demonstrate the relationship between systemic inflammatory biomarkers and anti-HCV.**

**Table 5. Threshold effect analysis of systemic inflammatory biomarkers on hepatitis C using two-piece wise linear regression.**

| Outcome (OR, 95%CI, *P* value) | Log PLR | Log SII | Log PMR | Log NMR |
|---|---|---|---|---|
| Model 1 Fitting model by standard linear regression | | | | |
| | 0.50 (0.42, 0.58) <0.0001 | 0.53 (0.47, 0.60) <0.0001 | 0.40 (0.34, 0.48) <0.0001 | 0.48 (0.40, 0.58) <0.0001 |
| Model 2 Fitting model by two-piece wise linear regression | | | | |
| Inflection point (K) | 7.62 | 9.11 | 9.3 | 3.55 |
| < K | 0.47 (0.40, 0.56) <0.0001 | 0.41 (0.36, 0.47) <0.0001 | 0.34 (0.29, 0.41) <0.0001 | 0.42 (0.35, 0.51) <0.0001 |
| > K | 2.19 (0.65, 7.36) 0.2034 | 1.65 (1.20, 2.26) 0.0020 | 2.79 (1.42, 5.49) 0.0030 | 3.99 (2.08, 7.63) <0.0001 |
| *P* for log likelyhood ratio test | 0.043 | <0.001 | <0.001 | <0.001 |

sensitivity of 0.5263; and NMR (AUC = 0.610) with a specificity of 0.7697 and sensitivity of 0.4515. Other markers were not included in the ROC analysis due to a lack of diagnostic value for HCV (Fig 3).

## 4 Discussion

Our pursuit in the exploration and application of inflammatory indicators remains ceaseless. We endeavor to identify an indicator capable of concurrently offering multiple indications of a disease in the realms of diagnosis, treatment, monitoring, and cure, with the potential for generalization in its use. In recent years, a growing body of research has emerged, employing ratios derived from routine blood tests to investigate infections, cancer, and various other diseases. Noteworthy studies have also been conducted in the context of viral hepatitis and virus-related HCC. In an investigation encompassing 189 consecutive adult liver transplant recipients with HBV related HCC, the findings indicated that elevated SII, PLR, and NLR represent markedly unfavorable prognostic indicators for both overall survival and recurrence-free survival in individuals undergoing liver transplantation for HBV related HCC [23]. However, an additional cross-sectional investigation utilizing data from NHANES 2017−2020 and involving a cohort of 6,792 adults revealed implications that elevated SII levels are associated with hepatic steatosis [24]. In a subsequent follow-up investigation involving 120 patients with HCV infection and 40 healthy controls, the outcomes revealed that both the HCV related cirrhosis group and the HCV related HCC group exhibited lower PLR values compared to the healthy control group. Furthermore, it was observed that the PLR in the HCV cleared group was notably higher when compared to both the HCV untreated group and the HCV uncleared group [14]. Thus, PLR is strongly associated with HCV severity and virologic response. This is highly consistent with the conclusion in this paper that patients with HCV infection have low levels of PLR and SII. In our current investigation, both SII and PLR exhibited commendable diagnostic and predictive efficacy (SII: AUC = 0.643; PLR: AUC = 0.623). Limited research has been conducted on PMR and NMR in the context of liver disease, as evidenced by a study published in 2023 on HBV-DeCi. This particular study was the first to identify PMR as an independent predictor for 30-day mortality in HBV-DeCi patients (AUC = 0.826; cutoff value: 118.62, sensitivity: 80.95%, specificity: 77.27%). Furthermore, the combination of PMR and the MELD score demonstrated an enhanced prognostic accuracy (AUC = 0.911; cutoff value: 0.05, specificity: 95.24%, sensitivity: 78.57%) [12]. Regarding NMR, there is a lack of research on liver disease, with studies

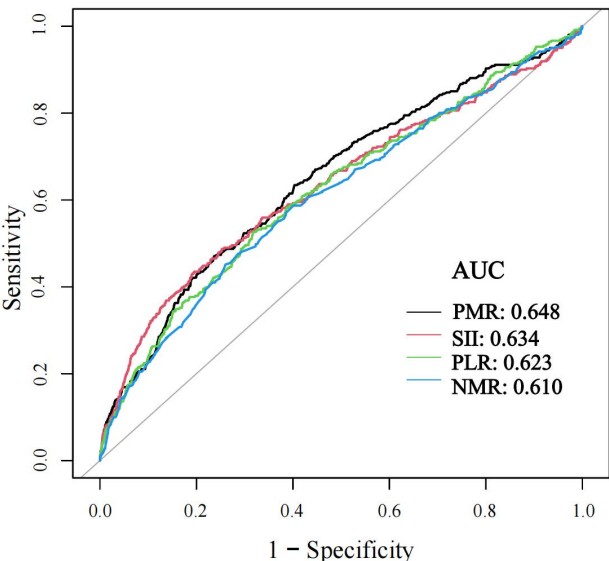

**Fig 3. Receiver operating characteristic curve analyses for the prediction of hepatitis C.**

focusing exclusively on severe SARS-CoV-2 infection (COVID-19) [25] and sepsis studies [26,27]. Therefore, our study firstly identified that PMR and NMR will as new accessible markers for HCV infection (AUC = 0.648 VS AUC = 0.610).

The findings of our study reveal that, in comparison to the uninfected group, various inflammatory biomarkers, including PLR, SII, PNR, PMR, and NMR, exhibited statistically significant reductions (OR < 1, Tables 2 and 3). This observation aligns with the typical profile associated with hepatitis-related viral infections, a consistent pattern noted in analogous studies [28,29]. The phenomenon is precisely opposite to the characteristics observed in certain respiratory viral infections, such as COVID-19 and influenza virus infection [30,31]. Hepatitis virus infection has a long course of disease, and long-term hepatitis virus infection can lead to hypersplenism and portal hypertension leading to thrombocytopenia, while the virus activates lymphocytes [32]. In the case of respiratory virus infection, because of its short course of illness, usually 5–7 days, during this process, platelets can be activated as inflammatory factors, and influenza virus infection can also lead to lymphocyte apoptosis and less circulating lymphocyte [33,34]. An additional noteworthy discovery underscores the nuanced interpretation of inflammatory ratios contingent upon demographic factors such as age, gender, and race. Notably, among individuals under 30 years of age, no statistically significant differences were observed in all CBC-derived inflammatory rates (Table 4; P > 0.05). Even within the cohort aged 30 and above, the optimal indicators for discerning infection persistently exhibit variations based on gender, with PNR and PMR favoring men, whereas PLR, SII, and NMR presenting greater diagnostic efficacy in women. Except for PMR, which exhibited significance across all racial groups in the NHANES database, other ratios with statistically significant differences were observed across races (Table 4). Research indicates that approximately half of patients with community-acquired acute hepatitis C can spontaneously clear the virus, and significantly higher levels of IL-1β, IL-1RA, IL-6, IFN-γ, and FGF-2 were detected in the serum of patients achieving self-limited infection [35,36]. Microbiota-modulated cytokines such as IL-1β, IL-18, and IL-22 can promote antiviral defense mechanisms by regulating adaptive immune responses. However, when the symbiotic state shifts toward pathology, these same antiviral cytokines paradoxically drive more severe disease progression to cirrhosis and hepatocellular carcinoma [37]. Routine cytokine profiling remains impractical for large-scale screening or asymptomatic subclinical populations. Therefore, Blood-cell-based inflammatory biomarkers — some studies showed were closely associated with cytokine networks — may provide cost-effective insights into antiviral potential and chronic infection progression [38,39].

Nevertheless, several limitations exist in this paper, and it is anticipated that these can be addressed through appropriate initiatives in subsequent studies: As a cross-sectional analysis based on NHANES data, our study relies on single-time sampling and survey data without access to longitudinal follow-up. Therefore, it cannot adequately address the potential value of continuous monitoring of these inflammatory markers. Based on the findings of this study, Blood-cell-based inflammatory biomarkers demonstrate no clinically significant correlation with HCV progression in individuals under 30 years of age. Consequently, there is insufficient evidence to support their utility for monitoring HCV progression in this demographic, necessitating continued reliance on alternative diagnostic modalities. To enhance this aspect, we intend to further validate and consolidate our conclusions in subsequent studies utilizing extensive clinical data from our laboratory. Simultaneously, the inflammatory markers in this study were derived from blood cell parameters, which exhibit a high degree of dependence on population, ethnicity, and lifestyle. It is our aspiration that the conclusions drawn from this US-based study will be applicable and translatable to other diverse populations.

## 5 Conclusion

Based on the comprehensive analysis presented in this study, the following conclusions were drawn. Firstly, in comparison to the group without HCV infection, certain inflammation rates, such as PLR, SII, PNR, PMR, and NMR, exhibited significant reductions. Secondly, all CBC-derived inflammatory ratios proved effective in diagnosing infections in individuals aged 30 and above, with no statistically significant differences observed in individuals under 30 years old. Lastly, regarding diagnostic significance, it was found that only PNR and PMR demonstrated a more favorable diagnostic outcome in

males compared to females, indicating lower values in males. Conversely, the diagnostic advantage of PLR, SII, and NMR indicators was more pronounced in females than in males, with lower values observed in the female population.

## Supporting information

**S1 Data. CBCHCV rawdata. This file contains the raw dataset used in the analysis of decreased levels of inflammatory markers in patients with hepatitis C virus (HCV) infection. The dataset includes variables such as demographic characteristics, clinical indicators, and measured inflammatory markers.**
(CSV)

## Acknowledgments

The authors express their gratitude to the staff members for their valuable contribution to data collection and for their efforts in making the data publicly accessible.

## Author contributions

**Conceptualization:** Lu Han, Lei Bi.

**Data curation:** Lu Han.

**Formal analysis:** Lu Han.

**Investigation:** Xinpeng Li.

**Methodology:** Xinpeng Li.

**Resources:** Xinpeng Li.

**Software:** Lei Bi, Xinpeng Li.

**Writing – original draft:** Lu Han, Xinpeng Li.

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
