## [Decision Letter · Decision Letter 0]

28 May 2025

Dear Dr. Li,

Thank you for submitting your manuscript to PLOS ONE. After careful consideration, we feel that it has merit but does not fully meet PLOS ONE’s publication criteria as it currently stands. Therefore, we invite you to submit a revised version of the manuscript that addresses the points raised during the review process.

The document contains spelling errors and requires grammatical editing. For example, in Table 1, the word 'unknow' should be corrected. A thorough revision is recommended.

The abbreviation ‘COVID-19’ is inconsistently written as both ‘COVID-19’ and ' covid-19' in different parts of the text. Please standardize its formatting

There are no page and line numbers used in the document- the authors did not follow PLOS instructions and makes it difficult to refer.

Please submit your revised manuscript by 28 June 2025, If you will need more time than this to complete your revisions, please reply to this message or contact the journal office at plosone@plos.org . Please include the following items when submitting your revised manuscript:

We look forward to receiving your revised manuscript.

Kind regards,

Maemu Petronella Gededzha, Ph.D

Academic Editor

PLOS ONE

Additional Editor Comments:

The document contains spelling errors and requires grammatical editing. For example, in Table 1, the word 'unknow' should be corrected. A thorough revision is recommended.

The abbreviation ‘COVID-19’ is inconsistently written as both ‘COVID-19’ and ' covid-19' in different parts of the text. Please standardize its formatting

There are no page and line numbers used in the document- the authors did not follow PLOS instructions and makes it difficult to refer

Reviewers' comments:

Reviewer's Responses to Questions

**Comments to the Author**

1. Is the manuscript technically sound, and do the data support the conclusions?

Reviewer #1: Yes

Reviewer #2: Partly

2. Has the statistical analysis been performed appropriately and rigorously?

Reviewer #1: Yes

Reviewer #2: Yes

3. Have the authors made all data underlying the findings in their manuscript fully available?

Reviewer #1: Yes

Reviewer #2: Yes

4. Is the manuscript presented in an intelligible fashion and written in standard English?

Reviewer #1: Yes

Reviewer #2: Yes

Reviewer #1: 2. Methods

2.1 HCV Diagnosis

There is a mention of molecular assays used by NHANES but not for anti-HCV (antibody-based) and the anti-HCV was used to categorize participants into anti-HCV pos or negative.

2.3 Study Population

Figure 1 and the paragraph above (that explains exclusion and sample size) on page 6 and 7, the numbers do not correlate – the figure states that “14007” participants enrolled (please correct grammar of enrolled (‘entolled’) in the last box of the figure) and the first sentence in pg 7 says 17 526 were enrolled. Please revise also the exclusion in the text and figure 1 for instance “blood metals” no where in the text is mentioned as exclusion criteria, as well as ≥18 years old on the excluded part-revise this as this impacts on age stratification. Revise numbers in the excluded in figure 1 in conjunction with those in the text above the figure 1.

2.5 Statistical Analysis

Page 8 12th row-…”predict sensitivity and specialist”.. it seems there is grammar error, was it not meant to be “specificity” instead of “specialist”

3. Results

3.1 Characteristics of participants (pg 8-9)

Results are based on total participants of 17 526, which is not in agreement with figure 1 which shows exclusion and enrollment of participants coming up to 14 007. Kindly address 2.3 of Methods to correctly reflect analyzed population number.

4. Discussion

Page 21 14th row- Please review and revise the sentence, grammar seems to be incomplete, sentence has no significance meaning as it written currently

Reviewer #2: The manuscript is well written, with detailed statistics however it would have been better had the authors detailed cytokines aligned to the ratios it would have given depth and a clearer understanding of the inflammatory mechanism in the study. The inability of the ratios to have any diagnostic value for those below 30 is a major limitation to recommend the use of the ratios for diagnostic purposes.

The OR are also not well represented and are difficult to interpret on current form. Also, although PLR, PMR and NMR show a negative association with HCV, this is not properly ventilated in the discussion.

If it is possible to address these minor comments, this is a well written and informative paper on HCV diagnosis.

**Do you want your identity to be public for this peer review?** For information about this choice, including consent withdrawal, please see our Privacy Policy

Reviewer #1: No

Reviewer #2: No

---

## [Author Response · Author response to Decision Letter 1]

22 Jul 2025

Manuscript ID: PONE-D-25-23527

Manuscript Title: The Diagnostic Value of Decreased Levels of Inflammatory Markers for the State of Hepatitis C Virus (HCV) Infection

Authors: Lu Han, Lei Bi, Xinpeng Li*

The authors appreciate the professional comments from the editors and reviewers. According to the comments, we have made extensive modifications to our manuscript to make our results convincing. All spelling errors have been corrected as the suggestions. The reference format has been revised to the “Vancouver” style as required by submission guidelines. Our point-by-point responses were presented as follows:

Response to Reviewer 1:

2.1 HCV Diagnosis

There is a mention of molecular assays used by NHANES but not for anti-HCV (antibody-based) and the anti-HCV was used to categorize participants into anti-HCV pos or negative.

The author’s answer: We have added specific details regarding HCV antibody testing in Methods 2.2, lines 104-116. The content is as follows:

HCV antibody was tested based on the VITROS Anti-HCV Reagent Pack and VITROS Immunodiagnostic Products Anti-HCV Calibrator on the VITROS ECi/ECiQ Immunodiagnostic Systems, the VITROS 3600 Immunodiagnostic System and the VITROS 5600 Integrated System. This involves a two-stage reaction. In the first stage, HCV antibody present in the sample binds with HCV recombinant antigens coated on the wells. Unbound sample is removed by washing. In the second stage, horseradish peroxidase (HRP)-labeled antibody conjugate (mouse monoclonal anti-human IgG) binds to any human IgG captured on the well in the first stage. Unbound conjugate is removed by washing. The system signal intensity is measured by chemiluminescence, and the signal intensity is directly proportional to the anti-HCV level in the sample.

2.3 Study Population

Figure 1 and the paragraph above (that explains exclusion and sample size) on page 6 and 7, the numbers do not correlate – the figure states that “14007” participants enrolled (please correct grammar of enrolled (‘entolled’) in the last box of the figure) and the first sentence in pg 7 says 17 526 were enrolled. Please revise also the exclusion in the text and figure 1 for instance “blood metals” no where in the text is mentioned as exclusion criteria, as well as ≥18 years old on the excluded part-revise this as this impacts on age stratification. Revise numbers in the excluded in figure 1 in conjunction with those in the text above the figure 1

The author’s answer: Thank you for your careful correction. We previously performed data adjustment for inclusion and exclusion, but we overlooked updating the data in the article's sections. We have now thoroughly reviewed and verified everything. The total number of participants from 2007 to 2018 is 59,842, and the final number of participants included in this study is 17,526. The corresponding spelling errors have also been corrected. Please refer to Figure 1 for details, with further information provided in lines 137-143.

2.5 Statistical Analysis

Page 8 12th row-…”predict sensitivity and specialist”.. it seems there is grammar error, was it not meant to be “specificity” instead of “specialist”

The author’s answer: The grammatical error has been corrected to "specificity," as detailed in line 172.

3. Results

3.1 Characteristics of participants (pg 8-9)

Results are based on total participants of 17 526, which is not in agreement with figure 1 which shows exclusion and enrollment of participants coming up to 14 007. Kindly address 2.3 of Methods to correctly reflect analyzed population number.

The author’s answer: This issue has been corrected, and the final number of participants included in this study is 17,526.

4.Discussion

Page 21 14th row- Please review and revise the sentence, grammar seems to be incomplete, sentence has no significance meaning as it written currently.

The author’s answer: The sentence indeed contained a grammatical error and has now been revised to: "As a cross-sectional analysis based on NHANES data, our study relies on single-time sampling and survey data without access to longitudinal follow-up. Therefore, it cannot adequately address the potential value of continuous monitoring of these inflammatory markers." Please refer to lines 338-342 for further details.

Response to Reviewer 2:

1.The manuscript is well written, with detailed statistics however it would have been better had the authors detailed cytokines aligned to the ratios it would have given depth and a clearer understanding of the inflammatory mechanism in the study. 

The author’s answer: We have reviewed some literature to clarify the relationship between the ratio, HCV, and cytokines in the discussion. Relevant content has been added to the discussion section in lines 325-336.

Research indicates that approximately half of patients with community-acquired acute hepatitis C can spontaneously clear the virus, and significantly higher levels of IL-1β, IL-1RA, IL-6, IFN-γ, and FGF-2 were detected in the serum of patients achieving self-limited infection. Microbiota-modulated cytokines such as IL-1β, IL-18, and IL-22 can promote antiviral defense mechanisms by regulating adaptive immune responses. However, when the symbiotic state shifts toward pathology, these same antiviral cytokines paradoxically drive more severe disease progression to cirrhosis and hepatocellular carcinoma. Routine cytokine profiling remains impractical for large-scale screening or asymptomatic subclinical populations. Therefore, Blood-cell-based inflammatory biomarkers — some studies showed were closely associated with cytokine networks — may provide cost-effective insights into antiviral potential and chronic infection progression.

2. The inability of the ratios to have any diagnostic value for those below 30 is a major limitation to recommend the use of the ratios for diagnostic purposes.

The author’s answer: Regarding the lack of diagnostic value of the ratio in the population under 30 years old in this study, we have added this point in the last paragraph of the discussion section, lines 342-346.

Based on the findings of this study, Blood-cell-based inflammatory biomarkers demonstrate no clinically significant correlation with HCV progression in individuals under 30 years of age. Consequently, there is insufficient evidence to support their utility for monitoring HCV progression in this demographic, necessitating continued reliance on alternative diagnostic modalities.

3. The OR are also not well represented and are difficult to interpret on current form. Also, although PLR, PMR and NMR show a negative association with HCV, this is not properly ventilated in the discussion.

The author’s answer: This negative association is primarily caused by thrombocytopenia resulting from hepatitis virus infection. We have added an explanation in the discussion section, lines 309-315.

Hepatitis virus infection has a long course of disease, and long-term hepatitis virus infection can lead to hypersplenism and portal hypertension leading to thrombocytopenia, while the virus activates lymphocytes. In the case of respiratory virus infection, because of its short course of illness, usually 5-7 days, during this process, platelets can be activated as inflammatory factors, and influenza virus infection can also lead to lymphocyte apoptosis and less circulating lymphocyte.

---

## [Decision Letter · Decision Letter 1]

27 Aug 2025

The Diagnostic Value of Decreased Levels of Inflammatory Markers for the State of Hepatitis C Virus (HCV) Infection

PONE-D-25-23527R1

Dear Dr. Xinpeng Li,

We’re pleased to inform you that your manuscript has been judged scientifically suitable for publication and will be formally accepted for publication once it meets all outstanding technical requirements.

Kind regards,

Maemu Petronella Gededzha, Ph.D

Academic Editor

PLOS ONE

Additional Editor Comments (optional):

Reviewers' comments:

Reviewer's Responses to Questions

**Comments to the Author**

Reviewer #1: All comments have been addressed

Reviewer #2: All comments have been addressed

2. Is the manuscript technically sound, and do the data support the conclusions?

Reviewer #1: Partly

Reviewer #2: Yes

3. Has the statistical analysis been performed appropriately and rigorously?

Reviewer #1: Yes

Reviewer #2: Yes

4. Have the authors made all data underlying the findings in their manuscript fully available?

Reviewer #1: Yes

Reviewer #2: Yes

5. Is the manuscript presented in an intelligible fashion and written in standard English?

Reviewer #1: Yes

Reviewer #2: Yes

Reviewer #1: (No Response)

Reviewer #2: The authors have addressed all the queries raised in the first review, I am happy to recommend this paper for publication.

**Do you want your identity to be public for this peer review?** For information about this choice, including consent withdrawal, please see our Privacy Policy

Reviewer #1: No

Reviewer #2: No

---

## [Editor Report · Acceptance letter]

PONE-D-25-23527R1

PLOS ONE

Dear Dr. Li,

I'm pleased to inform you that your manuscript has been deemed suitable for publication in PLOS ONE. Congratulations! Your manuscript is now being handed over to our production team.

Kind regards,

on behalf of

Dr. Maemu Petronella Gededzha

Academic Editor

PLOS ONE